# Consumption of Cooked Black Beans Stimulates a Cluster of Some Clostridia Class Bacteria Decreasing Inflammatory Response and Improving Insulin Sensitivity

**DOI:** 10.3390/nu12041182

**Published:** 2020-04-23

**Authors:** Mónica Sánchez-Tapia, Irma Hernández-Velázquez, Edgar Pichardo-Ontiveros, Omar Granados-Portillo, Amanda Gálvez, Armando R Tovar, Nimbe Torres

**Affiliations:** 1Departamento de Fisiología de la Nutrición, Instituto Nacional de Ciencias Médicas y Nutrición Salvador Zubirán, Mexico City 14080, Mexico; qfbmoniktc@gmail.com (M.S.-T.); edgar.pichardo@gmail.com (E.P.-O.); ograpo@yahoo.com (O.G.-P.); armando.tovarp@incmnsz.mx (A.R.T.); 2Facultad de Química, Universidad Nacional Autónoma de México, Mexico City 0410, Mexico; hernandezvelazquezia@gmail.com (I.H.-V.); galvez@unam.mx (A.G.)

**Keywords:** black bean, gut microbiota, *Ruminococcus bromii*, resistant starch, incretins, SCFA, insulin sensitivity

## Abstract

There is limited information on the effect of black beans (BB) as a source of protein and resistant starch on the intestinal microbiota. The purpose of the present work was to study the effect of cooked black beans with and without high fat and sugar (HF + S) in the diet on body composition, energy expenditure, gut microbiota, short-chain fatty acids, NF-κB, occluding and insulin signaling in a rat model and the area under the curve for glucose, insulin and incretins in healthy subjects. The consumption of BB reduced the percentage of body fat, the area under the curve of glucose, serum leptin, LPS, glucose and insulin concentrations and increased energy expenditure even in the presence of HF + S. These results could be mediated in part by modification of the gut microbiota, by increasing a cluster of bacteria in the Clostridia class, mainly *R. bromii*, *C. eutactus*, *R. callidus*, *R. flavefaciens* and *B. pullicaecorum* and by an increase in the concentration of fecal butyrate. In conclusion, the consumption of BB can be recommended to prevent insulin resistance and metabolic endotoxemia by modifying the gut microbiota. Finally, the groups fed BB showed lower abundance of hepatic FMO-3, even with a high-fat diet protecting against the production of TMAO and obesity.

## 1. Introduction

Diet can influence the composition of the human microbiota; however, there is limited information on the effect of legumes as a source of protein and, because of their content of resistant starch content, with respect to their impact on the intestinal microbiota. Legumes or pulses are an important source of protein and complex carbohydrates, that serves as a major source of energy in the human diet. The carbohydrate content in pulses is 60–65%, with starch constituting the largest fraction of the carbohydrates. The two components of starch are amylose, which is a linear molecule, and amylopectin, which is highly branched, making amylopectin easily accessible to digestive enzymes in the small intestine. Resistant starch (RS) contains α-linked glucose molecules that are resistant to hydrolysis in the small intestine, passing directly through the colon where it is fermented by the resident microbiota. RS are classified into four groups based on the properties that allow it to resist digestion. RS type 1 (RS1) is found in whole grains and legumes and is inaccessible to digestive enzymes because it is surrounded by a protective matrix. RS type 2 (RS2) contains dense ungelatinized starch granules that are resistant to digestion before cooking. RS2 includes unripe bananas, uncooked potatoes and high-amylose maize starch. RS3 are retrograded starches that are formed when starchy foods are cooked and then cooled. RS type 4 (RS4) is formed via chemical cross-linking of starch by the addition of esters and ether groups. RS4 are usually developed for use in processed foods [1,2]. Upon entering the colon, RS undergoes a high degree of anaerobic fermentation by the local microbiota into hydrogen, methane, carbon dioxide and short-chain fatty acids (acetate, propionate, and butyrate). Butyrate is the predominant short-chain fatty acid produced from RS, which is involved in maintaining intestinal homeostasis. Butyrate has several proposed health benefits, including providing energy for colonic epithelial cells and improving insulin sensitivity. The best natural sources of RS are less-mature bananas, pulses and potatoes. Estimates indicate that intake of pulses is low in developed countries, approximately 3 g per day, whereas in developing countries, intakes may be as high as 30–40 g per day [3]. The United Nations declared 2016 to be the “Year of the Pulses” in order to raise awareness, and position pulses as a primary source of protein and therefore of RS [4]. BBs are unusual in their starch content, because a large portion of their starch comes in the form of resistant starch. Since this breakdown of RS does not occur in the small intestine, the starch is not converted into simple sugars avoiding a quick rise in blood sugar level. Therefore, RS may be used as a substrate in the colon by the gut microbiota. Although isolated RS is known to have health benefits, the association between BB and improvements in glucose and lipid metabolism with changes in the gut microbiota is unclear.

In addition to our understanding of the role of BB in specific changes in the gut microbiota, limited research has been done on the possible beneficial effects of the consuming of BB as a vegetable protein and source of RS to prevent development of insulin resistance in comparison with animal protein. In addition, more human research is needed to study the effect of BB as an aid to glycemic control. Thus, this work was undertaken to obtain a mechanistic understanding of how BB consumption might modify the gut microbiota and what the metabolic consequences are. To address this question, a combination of molecular approaches were used to determine the composition of the gut microbiota after the consumption of dried and cooked BB, and their effects on energy expenditure, body composition, glucose tolerance and insulin signaling pathway and metabolic endotoxemia.

## 2. Materials and Methods

### 2.1. Animals

The purpose of this study was (1) to determine the effect of consumption of dry cooked black bean (BB) in a control or a high-fat diet with 5% sucrose in the drinking water (BB + HF + S) compared to those fed casein (C) in a control or high-fat diet with 5% sucrose in the drinking water (C + HF + S) on the gut microbiota, to study the effect of the type of protein and the presence of resistant starch in the BB, and (2) to study the effect of an acute metabolic challenge with glucose after the consumption of BB in rats fed control or high-fat diet on insulin signaling and G-protein-coupled receptors GPR43 and GPR41 and hepatic flavin monooxygenase 3 (FM03) protein abundance, to determine the protective effect of long-term consumption of a vegetable protein, such as BB, with resistant starch compared to an animal protein, such as casein, on insulin resistance.

Study 1. Six-week-old male Wistar rats were obtained from the National Institute of Medical Sciences and Nutrition. The animals were housed in individual cages and maintained at a controlled room temperature with 12-h light-dark cycles and free access to water and food. Rats were randomly assigned to four groups; eight rats per group were assigned to one of four experimental diets: (1) 17% casein (control group); (2) 17% black bean protein from dry cooked black bean (BB); (3) a casein high-fat diet with 5% sucrose in drinking water (C + HF + S); and (4) a BB high-fat diet with 5% sucrose in the drinking water (BB + HF + S), Table 1. The four groups were fed for two months. Animal weight and food consumption were determined two days per week during the protocol. At the end of the study, stools of each rat were collected and stored at −70 °C until analysis. 

Study 2. Before euthanasia of the four groups mentioned above, each experimental group (*n* = 8) was divided into two subgroups; the first subgroup was killed in the fasting stage (*n* = 4) and the other subgroup (*n* = 4) had a metabolic challenge of one dose of glucose (2 g per kg of body weight) by intraperitoneal administration 30 min before euthanasia. At the end of the experiment, the rats were killed by decapitation after being anesthetized with CO_2_. The liver, intestine and muscle was rapidly removed and stored at −70 °C until analysis. The serum was obtained by centrifugation of the blood at 1500× *g* for 10 min and stored at −70 °C until further analysis. The Animal Committee of the National Institute of Medical Sciences and Nutrition, Mexico City (CINVA1744) approved the procedure.

### 2.2. Biochemical Parameters

Serum biochemical parameters, including glucose, triglycerides, total, and LDL cholesterol were analyzed with a COBAS C111 (Roche, Basel, Switzerland). Serum insulin (Alpco Diagnostics, Salem, NH, USA) and LPS (Cloud-Clone Corp, Houston, TX, USA) were determined using commercial ELISA kits.

### 2.3. Glucose Tolerance Test

The glucose tolerance test was determined by the administration of an intraperitoneal injection of a glucose load of 2 g per kg body weight in fasted rats. Blood samples were collected from the tail vein at 0, 15, 30, 45, 60, 90, and 120 min after administration of the glucose. Plasma glucose concentration was measured using a FreeStyle Optium glucometer (Abbot Laboratories, AbbotPark, IL, USA)

### 2.4. Energy Expenditure

The energy expenditure was determined by indirect calorimetry in an Oxymax Lab Animal Monitoring (CLAMS) System (Columbus Instruments, Columbus, OH, USA). The animals were individually housed in plexiglass cages with an open-flow system connected to the CLAMS. Throughout the test, O_2_ consumption (VO_2_ mL/kg/h) and CO_2_ production (VCO_2_, mL/kg/h) were measured sequentially for 90 s. The respiratory exchange ratio (RER) was calculated as the average ratio of the produced VCO_2_ produced to the inhaled VO_2_ (VCO_2_/VO_2_).

### 2.5. Fecal DNA Extraction and 16S rRNA Sequencing

A fecal sample was collected from all animals after 2 months of the dietary treatment with black bean. Fecal samples were frozen at −80°C. DNA extraction was carried out using a QIAamp DNA Stool Mini Kit (Qiagen, U.S.A.), according to the manufacturer’s instructions. The variable regions 3–4 of the 16S rRNA gene were amplified using specific forward (5′ TCGTCGGCAGCGTCAGATGTGTATAAGAGACAGCCTACGGGNGGCWGCAG 3′) and reverse primers (5′ GTCTCGTGGGCTCGGAGATGTGTATAAGAGACAGGACTACHVGGGTATCTAATCC 3′) containing the Illumina adapter overhang nucleotide sequences. Ampure XP bits were used to purify the 16S V3-V4 amplicons, and were quantified on Qiaxcel (QIAGEN, Germany) The size of amplicons was approximately 550 bp. An index PCR was then carried out to attach dual indices using a Nextera XT v2 Kit. The amplicon size was approximately 610 bp, and the concentration of double-stranded DNA was measured using a fluorometer Qubit 3.0 with a high-sensitivity kit. The final amplicon library was pooled in equimolar concentrations. Sequencing was performed on the Illumina MiSeq platform (MiSeq Reagent Kit V.3, 600 cycles) at 15 pM with 20% Phyx infection according to the manufacturer’s instructions to generate paired-end reads of 300 bases in length in each direction. 

### 2.6. Bioinformatic Analysis

For taxonomic composition analysis, Custom C# and python scripts in the Quantitative Insights Into Microbial Ecology (QIIME) software pipeline 1.9 were used to process the sequencing files [5]. The sequence outputs were filtered for low-quality sequences (defined as any sequences that are <200 bp or >620 bp, sequences with any nucleotide mismatches to either the barcode or the primer, sequences with an average quality score of <30, and sequences with ambiguous bases >0). Sequences were checked for chimaeras with Gold.fa, and chimeric sequences were filtered out. The analysis started by clustering sequences within a percent sequence similarity into operational taxonomic units (OTUs). Ninety-five percent of the sequences passed filtering, resulting in 52,974 ± 16,571 sequences/sample on average, with a 97% similarity threshold. OTU picking was performed with the tool set from QIIME, using the Usearch method [6]. OTUs were picked against the GreenGenes v.13.9 database. Ninety-seven percent of the OTUs were selected from the database. After the resulting OTU files were merged into one overall table, taxonomy was assigned based upon the GreenGenes reference taxonomy database. Thus, 99.4%, 99.2%, 99.2%, 91.9%, 78.4% and 34.4% of the reads were assigned to the phylum, class, order, family, genus and species level, respectively. Species richness (Observed, Chao1) and alpha diversity measurements (Shannon) were calculated, and we estimated the within-sample diversity at a rarefaction depth of >15,270 reads per sample. Weighted and unweighted UniFrac distances were used to perform the principal coordinate analysis (PCoA). Microbial sequence data were pooled for OTU comparison and taxonomic abundance analysis but separated by batch in the principle coordinates analysis (PCoA) to have clear PCoA figures. For even sampling, a depth of 15,270 sequences/sample was used. PCoAs were produced using Emperor, and the community diversity was determined by the number of OTUs and beta diversity, measured by UniFrac unweighted and weighted distance matrices in QIIME. ANOSIM, a permutational multivariate analysis of variance, was used to determine statistically significant clusterings of groups based upon microbiota structure distances.

### 2.7. Western Blot Analysis

The total protein of the individual samples (*n* = 4) was extracted and quantified by the Bradford assay (Bio-Rad, Hercules, CA, USA) and stored at −70 °C. The protein detection was performed by electrophoresis in SDS-PAGE and then transferred to polyvinylidene difluoride membranes. Blots were blocked with 3% BSA for 60 min at room temperature and incubated overnight at 4 °C with the primary antibodies. The blots were incubated with anti-rabbit, anti-goat, or anti-mouse secondary antibodies conjugated with horseradish peroxidase (1:15,000). GAPDH (1:3500) was used to normalize the data. The images were analyzed with a ChemiDocTM XRS + System Image LabTM Software (Bio-Rad, Hercules, CA, USA). The assays were performed three times using independent blots.

### 2.8. Glycemic Index in Healthy Subjects

Eight healthy, nonsmoking, nonmedicated, and normal-weight volunteers (7 women and 1 man, mean age 27.1 ± 2.6 years, BMI 18.9 ± 7.8 kg/m^2^) were included in this study. Each participant arrived in the morning after a 10-h overnight fast and consumed the reference meal (25 g of glucose) three times on separate days, and the test meal was only consumed once. Blood samples were obtained before and at 15, 30, 45, 60, 90 and 120 min after the test meals. At each time, the serum glucose concentration was determined by enzymatic methods (Roche, COBAS C111, Rotkreuz, Switzerland). Serum Insulin, glucagon like peptide-1 (GLP-1) and plasma glucose-dependent insulinotropic polypeptide (GIP) concentration was obtained by ELISA kits (Millipore, Billerica MA, USA). The study was approved by the Institutional Committee for Human Biomedical Research (no. 2991) of the INCMNSZ.

### 2.9. Statistical Analysis

The results were expressed as mean ± SEM. Statistical analysis was performed using two-way ANOVA followed by Bonferroni’s post-hoc test, using Prism 7.0 software (GraphPad, San Diego, CA, USA); *p* < 0.05 was considered significant.

## 3. Results

### 3.1. Black Bean Maintains Lean Mass and Reduces Body Fat by Increasing Energy Expenditure 

After two months of dietary treatment with different types of protein, those fed with C gained significantly (32.9%) more weight than those fed BB protein. Interestingly, rats fed BB + HF + S gained significantly less weight (32.2%) than those fed C + HF + S (Figure 1A). When body composition was measured, rats fed BB protein showed significantly less % body fat (−28.5%) than rats fed the C diet, and, remarkably, rats fed BB + HF + S showed the same percentage of body fat as the BB group (Figure 1B). To demonstrate whether the changes in body weight gain were due to the protein quality, which could be reflected in the % of lean body mass, this was measured in the four groups. Contrary to our expectations, rats fed BB, BB + HF + S or C showed similar % lean body mass (74%) and rats fed C + HF + S showed significantly lower % lean body mass than the C group (−12.5%), indicating that although the protein quality is not the same in both proteins, these differences mainly affect body fat content (Figure 1C).

Since the type of protein and the presence of a high-fat diet with sugar (HF + S) had differential effects on weight gain and body composition, we further studied energy expenditure by indirect calorimetry to assess whether there was an association between the type of protein, body composition, body weight and energy expenditure. The C and BB groups had similar VO_2_ consumption, indicating similar energy expenditure (Figure 1D); however, the C group showed the highest body weight and body fat gain, suggesting that energy expenditure is more associated with the % of lean body mass (*r* = 0.85, *p* < 0.0001), (Figure 1E). Therefore, the respiratory exchange ratio (RER) was measured to evaluate the type of energy substrate used for energy expenditure. A low RER (0.70) reflects, predominantly, fat oxidation, whereas a high RER (1.00) is indicative of glucose oxidation. During the fasting state, all groups showed an RER of approximately 0.7, indicating that the utilized substrate was fat. Interestingly, the C and the BB groups were characterized by periodic shifts in glucose and fat oxidation during fasting and refeeding. However, the group fed C + HF + S, with the highest body fat mass, showed a RER of 0.75 during the fasting and refeeding period, indicating a metabolic inflexibility caused by impaired fuel switching and a state of metabolic insensitivity and inflexibility. However, the group fed BB + HF + S showed partial metabolic inflexibility (Figure 1F).

### 3.2. Black Bean Down Regulate Leptin and Insulin Concentration Decreasing Glucose Intolerance

It is not known whether BB can reduce the glycemic response due to the type of protein and the presence of resistant starch. Therefore, we carried out a glucose tolerance test in rats fed the different proteins. Rats fed C + HF + S showed a significant increase in the area under the curve for glucose (AUC) with respect to the C group (85%). Interestingly, BB or BB + HF + S showed even lower AUC than the C group, indicating that the presence of RS in BB resists hydrolysis in the small intestine, passing directly through the colon, where it is fermented by the resident microbiota avoiding the postprandial peaks of glucose (Figure 2A,B). These results were reflected in the concentrations of glucose and insulin. Groups fed with BB, even in the presence of HF + S, showed the lowest concentration of glucose and insulin compared to the C group (Figure 2C,D). Due to the low body fat content in the rats fed BB or BB + HF + S, we measured the concentration of serum leptin. Interestingly, rats fed with BB or BB + HF + S showed the lowest concentration of leptin than either the C group or the C + HF + S group (Figure 2E). There was a significant correlation between leptin concentration and body fat (Figure 2F).

### 3.3. Black Bean Modifies Gut Microbiota by Increasing a Cluster of Bacterias of the Clostridia Family and Decreasing Metabolic Endotoxemia and Glucose Concentration

Purified resistant starch has been shown to modify the gut microbiota; however, the resistant starch contained in black beans is the way different populations consume resistant starch. Therefore, we evaluated the effect of black bean on gut microbiota. Rats fed BB + HF + S followed by BB showed higher α-diversity than rats fed C or C + HF + S (Figure 3A). The PCoA analysis revealed that 72.7% of the variation in the gut microbiota can be explained by the type of protein and 18.0% by the presence of a high content of fat and sucrose in the diet. The weighted beta diversity analysis showed that the microbial communities were significantly different in the four groups (ANOSIM *r* = 0.999, *p* = 0.001), (Figure 3B). The results of the sequencing showed that, at the phylum level, the groups fed BB increased cyanobacteria and decreased proteobacteria with respect to the C group. The HF + S groups, regardless of the type of protein, increased the proteobacteria. The C + HF + S group showed a significant increase in Lentisphaerae phylum (Figure 3C). At the class level, the groups fed with BB showed a significant increase in the Clostridia class (Figure 3D). At genus level, the groups fed BB showed the highest abundance of *Ruminococcus*, *Coprococcus* and *Prevotella* relative to the rest of the groups. The presence of HF + S in the diet increased *Bacteroides* (Figure 3E); however, interestingly, rats fed with C + HF + S significantly decreased *Ruminococcus* compared to BB + HF + S. At the species level, 24% of the reads were assigned with 13,000 reads per sample. The most abundant bacteria at species level in the BB group were *R. bromii*, *C. eutactus*, *R. callidus*, *R. flavefaciens* and *B. pullicaecorum*, most of them butyrate producers. In the HF groups, rats fed BB + HF + S maintained the presence of *B. pullicaecorum* and *R. callidus* (Figure 3F). As a consequence, we showed that the BB groups had the lowest concentrations of circulating LPS (Figure 3G). In addition, we observed a significant inverse correlation between the abundace of R. bromii and serum glucose concentration. 

### 3.4. Consumption of Black Bean Increases Butyrate and Decreases NF-κB Protein Abundance in Colon

The dietary resistant starch contained in the black bean is often the major source of energy contributing to the growth of *Ruminicoccus bromii*, due to the fermentation of RS by this bacteria in the colon. Therefore, we were interested in measuring the concentration of fecal short-chain fatty acids (SCFA). The highest concentration of total SCFA was observed in the group fed C + HF + S (88.7 mmol/g feces), followed by the C group (77.1 mmol/g feces), BB (65.8 mmol/g feces) and the BB + HF + S group (61.9 mmol/g feces) (Appendix A). There was no change in the concentration of acetate in all groups; however, the highest concentration of propionate was observed in the group fed C + HF + S (Appendix A) and the lowest concentration was observed in the groups fed BB even in the presence of HF + S in the diet. Interestingly, the group fed BB showed the highest concentration of butyrate, followed by the group fed BB + HF + S, and the groups fed C or C + HF + S showed the lowest concentration (Figure 4A).

SCFAs bind to the receptors GPR43 and GPR41 in the intestine. Thus, we measured the protein abundance of these receptors in the intestine. GPR43 has more affinity for acetate, while GPR41 has more affinity for butyrate. In the present study, we observed a 2-fold increase in GPR43 in the group fed C + HF + S compared to the rest of the groups (Figure 4B,C) and there was no change in the protein abundance of GPR41 in all groups (Figure 4B,D).

The functions of SCFAs and their receptors are involved in regulation of immune response including the gut barrier function. Previous studies have demonstrated that butyrate improves the function of the intestinal epithelial barrier [7]; thus, we measured the abundance of occludins in the colon. Interestingly, the groups fed BB showed the highest abundance of occludins, whereas the group fed C + HF + S showed the lowest abundance of occludins. There was a significant possitive correlation between occludin abundance and butyrate concentration (Figure 4 F). Because butyrate deficiency has been associated with modulation of the transcription factor NF-κB involved in the regulation of cytokines [8], we measured the intestinal abundance of NF-κB. The group fed with BB showed the highest butyrate concentration and the lowest abundance of NF-κB (Figure 4B,G) followed by the group fed BB + HF + S, whereas the group fed with C + HF + S showed the highest abundance of this transcription factor (Figure 4B,G).

The high-fat diet increased the concentration of LPS associated with high intestinal permeability. In the present study, animals fed C + HF + S showed a significant increase in serum LPS concentration (Figure 3G) associated with a lower concentration of butyrate and occludins and a higher abundance of NF-κB and GPR43 in the intestine, whereas groups fed BB even in the presence of HF + S showed the lowest LPS concentration. There was a positive significant correlation between occludin and butyrate (Figure 4F) and LPS and GPR-43 (Figure 4H) and a negative association between NF-κB and butyrate (Figure 4G).

### 3.5. The Low Glycemic Index of the Black Bean Modulate GIP Secretion and Increase Insulin Sensitivity

Therefore, we next sought to investigate whether an acute exposure of glucose had any impact on insulin sensitivity after the consumption of black beans in rats. After 2 months of different dietary treatments, four rats from each group received 2 g/kg weight of glucose and were killed 30 min later. Remarkably, after 30 min of this challenge, rats fed with BB even in the presence of a HF diet showed a significant increase in GPR41 and occludin abundance, indicative of butyrate production and adequate epithelial barrier function, and this was not observed in rats fed C or C + HF + S (Figure 5A). The abundance of GIP during the fasting state (Figure 4B) was reduced in all four groups; however, after the glucose stimulation, GLP-1 was induced in the C, BB and BB + HF + S groups (Figure 5B), indicating that butyrate produced by the RS of BB binds to GPR41, increasing GLP-1 secretion by L cells. GIP was stimulated in all groups, with the lowest abundance in the BB group. Long-term consumption of BB increased IRS-Tyr phosphorylation and increased protein kinasee B (AKT) phosphorylation, indicating an increasing sensitivity to insulin even in the presence of HF (Figure 5C). This was confirmed in eight healthy subjects that consumed 25 g of available carbohydrates (166 g of cooked BB). The area under the curve (AUC) for glucose after the consumption of BB was significantly lower than after the consumption of 25 g of glucose, and the glycaemic index of BB was 46.6 (Figure 6A). The AUC for insulin was also significantly lower after BB consumption, and the insulinemic index was 59.3 (Figure 6B). The AUCs for GIP and GLP were similar in the BB and glucose groups (Figure 6C,D).

Finally, It has been demonstrated that omnivorous subjects produce significantly more trimetyhylamine (TMA) than vegetarians by the gut microbiota [9], which is further metabolized into proatherogenic species, particularly trimethylamine-N-oxide (TMAO) by the flavin monoxygenase 3 (FMO3) in the liver. Therefore, we analyzed the FMO3 in the liver. Interestingly, FMO3 was inducible by C and more evident by the C + HF + S group; however, FMO3 abundance was barely expressed after the consumption of BB, even with HF + S (Figure 5D), suggesting that BB consumption may reduce the atherogenic capacity of a high-fat diet and that low abundance of FMO3 in the BB group was associated with lower body weight gain and lower % of body fat (Appendix A).

## 4. Discussion

The pulse or legumes are an important source of nutrition for billions of people around the world. Beans contain approximately 20–25% protein with a protein quality of about 60% [10], and the limiting amino acids in these pulses are methionine, tryptophan and cystine. The staple foods in some countries are pulses in combination with cereals to meet their protein requirement and protein synthesis [11]. The group fed BB gained less weight than the group fed casein, this could be due to the quality of the protein; however, interestingly, when body composition was measured, the groups fed BB even with a high-fat diet showed significantly less % body fat than in the C group. The BB group showed the same lean mass as the C group, indicating that BB is a good source of protein and the lower weight gain could be due to the lower % of body fat. Studies with white kidney bean have reported a reduction in fat accumulation [12]. The increase in fat mass has been associated with low oxygen consumption [13]; in the present work, we observed a significant increase (22.5%) in VO_2_ in the groups fed BB and less body fat with respect to the C group, indicating that BB stimulates energy expenditure. In fact, there was a significant correlation between the % of lean mass and VO_2_ consumption. A possible explanation for these findings could be the changes in gut microbiota, mainly in butyrate-producing bacteria, as has been described [14,15]. Furthermore, one of the main components of beans is resistant starch, and, although there is considerable information on the effect of the different types of resistant starch on gut microbiota [16], few studies have focused on the effect of beans containing naturally resistant starch on the intestinal microbiota. One of the major sources of energy for microbial growth in the human colon is dietary resistant starch, which escapes digestion in the small intestine and reaches the colon undigested. Fermentation of these substrates provides SCFA, mainly butyrate [17], which is beneficial to the health of the colon as energy source [18] and for the maintenance of gut barrier function through the increase in occludins [19,20]. Occludins are important proteins to maintain the stability of tight junction and barrier function [20]. In addition, we observed that the groups fed BB inhibited the activity of the transcription factor NF-κB, which is involved in the regulation of pro-inflammatory cytokines [8]. Short-chain fatty acids, in particular butyrate, bind to the GPR41 receptor expressed in the colon, whereas acetate and propionate bind mainly to GPR43 [21]. Interestingly, the group fed C + HF + S induced the protein abundance of GPR43, which was associated with higher serum glucose, insulin, leptin and glucose intolerance than the C group, whereas consumption of BB or BB + HF + S showed an opposite pattern with a lower concentration of the biochemical parameters described above, even lower than the C group associated with a lower GPR43 protein abundance. Interestingly, acetate and propionate stimulate adipogenesis via GPR43 [22], and inhibition of GPR43 reduces fat mass accumulation [23]. 

The evidence suggests that SCFA may play a role in signaling satiety by providing a possible mechanism underlying the effect of RS consumption and appetite control. SCFA activates L-cells to secrete glucagon-like peptide (GLP-1). GLP-1 in intestine increased in the groups fed BB, followed by BB + HF + S and C groups, and GLP-1 abundance was almost suppressed by the C + HF + S, indicating that consumption of BB could increase satiety, as has been demonstrated in humans [24].

Interestingly, we found that the consumption of BB increases alpha diversity and modifies the gut microbiota by increasing the diversity of bacterial communities, mainly of the major phyla Gram-positive Firmicutes and of the class Clostridia of the genus *Ruminococcus*, mainly *R. bromii*, a key species for the degradation of resistant starch in the human colon [16], *R. callidus* and *R. flavefaciens*. From the genus *Butyriciccocus B. pullicaecorum* mainly increased, and from the *genus Coprococcus*, mainly *C. eutactus*, key species for the degradation of resistant starch in the human colon [25]. In addition, the genus *Coprococcus* is a net acetate producer and has detectable butyrate kinase, acetate kinase, and butyryl-CoA:acetate-CoA transferase activities to produce butyrate [26]. The presence of *B. pullicaecorum* has been reported to decrease inflammation in a rat colitis model, prevent cytokine-induced epithelial integrity losses [27] and is capable of producing a high concentration of butyrate. We demonstrated that the *B. pullicaecorum* was present in the group fed black bean, even in the presence of a high-fat diet. Genetic analyses have shown that *R. bromii* has a unique and specialized organization of extracellular amylases that give it the ability to ferment RS, suggesting that it may be a crucial species for RS degradation [28]. The group fed BB had approximately 8-fold increased *R. bromii* relative to the C group; however, the group fed BB + HF + S had decreased *R. bromii*, by 27%. It has been reported that subjects with obesity have a decreased concentration of butyrate and butyrate-producing bacteria in their stool [29]. The main product of *R. bromii* fermentation in the large intestine is butyrate, which has a particularly important role as the preferred energy source for the colonic epithelium. 

On the other hand, it is known that, although BB has some limiting amino acids, recent studies have shown that *R. bromii* increases the induction of the tryptophan biosynthesis genes [16], and methionine [30], the limiting amino acids of BB. In addition to the fermentation of indigestible carbohydrates, gut microbes are able to convert dietary choline and L-carnitine, via microbial enzymes, into trimethylamine (TMA). TMA can be transported via the portal vein to the liver, where it can further be metabolized into trimethylamine N-oxide (TMAO) by the liver enzyme flavin monooxygenase (FMO)-3. Omnivores were more likely to produce a high concentration of TMAO than vegetarians [9], because their microbiota produce a higher concentration of TMA and also have higher expression of FMO3 [31]. Interestingly, the groups fed BB showed significantly lower hepatic FMO-3 abundance, indicating that black bean consumption, even with a high-fat diet, protects against the production of TMAO. Previous studies reveal a link between reduced FMO3 expression and protection against obesity [32]. In fact, we observed an association between low FMO3 abundance with a low % of body fat. In contrast, the casein-fed groups showed a significantly higher hepatic FMO-3 abundance, and the addition of fat in the diet further increased FMO-3, % body fat, and high insulin levels, which have been associated with increased production of TMAO, obesity and insulin resistance [32]. The principal component analysis of the gut microbiota indicated that 72.7% of the variation in gut microbiota is explained by the type of protein and 18.0% of the variation is explained by the presence of a high-fat diet. The PC analysis showed that the microbiota of the group fed BB is different from that of the group fed C, indicating that the plant-based diet, in particular the black bean protein diet, induces significant differences in the gut microbiota. 

Resistant starch has been considered as a prebiotic and as a dietary strategy for the complications of obesity by reducing hyperinsulinemia and increasing the concentration of HDL-C [33], and by reducing fat accumulation. In fact, the group fed BB showed 26.6% less body fat than the group fed C, whereas the group fed BB + HF + S showed 57.7% less body fat than the group fed with C + HF + S. Interestingly, the groups fed with BB showed lower concentrations of serum leptin, insulin, glucose concentration and AUC after a glucose tolerance test than the group fed C, avoiding the postprandial glucose peaks. These results may be explained in part by a better response of the insulin pathway after a challenge with glucose in skeletal muscle, preventing insulin resistance. These results were confirmed in healthy subjects, who showed lower area under the curve for glucose and insulin after black bean consumption. 

In addition, beans can provide phenolic compounds with antioxidant activity and phytosterols. Phytosterols and phenolic compounds from the black bean seed coat can potentially reduce cholesterol levels via the gut microbiota [34], by inhibiting the micellar solubility of cholesterol and reducing the expression of lipogenic genes [35,36]. A link has been suggested between the consumption of foods rich in polyphenols and the reduction of the incidence of numerous chronic disorders; however, inside the human body, the chemical structure of most polyphenols is received as xenobiotic and, therefore, the bioavailability of these compounds is greatly reduced. Because of their poor absorption (about 5–10%), the remaining polyphenols (90–95%) are retained in the large intestine for a longer time with the bile conjugates released into the lumen and are exposed to microbial enzymatic activities in the gut, exhibiting prebiotic effects and antimicrobial action against pathogenic intestinal microbiota [37].

## 5. Conclusions

Our study demonstrated that consumption of black beans improved glucose response, an effect mediated in part by modification of the gut microbiota, by increasing a cluster of Clostridia class bacteria. This group of bacteria seems to be a good candidate that for probiotic use, with anti-inflammatory potential. The modification of the gut microbiota by BB is, in part, associated with the presence of resistant starch. The use of BB can be recommended a part of the dietary strategy to treat subjects with insulin resistance.

## Figures and Tables

**Figure 1 nutrients-12-01182-f001:**
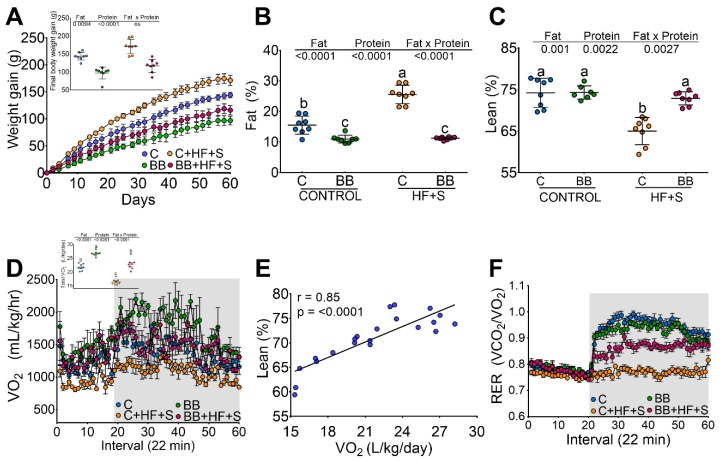
Black bean maintains lean mass and reduces body fat by increasing energy expenditure. (**A**) Weight gain after the consumption of casein (**C**), black bean (BB), C + high-fat + 5% sucrose (HF + S) (C + HF + S), BB + HF + S for 2 months. Inset: final body weight. (**B**), body fat, (**C**), lean body mass. (**D**) oxygen consumption, (**E**) correlation between lean mass and oxygen consumption, (**F**) respiratory exchange ratio (RER), *n* = 8 per group.

**Figure 2 nutrients-12-01182-f002:**
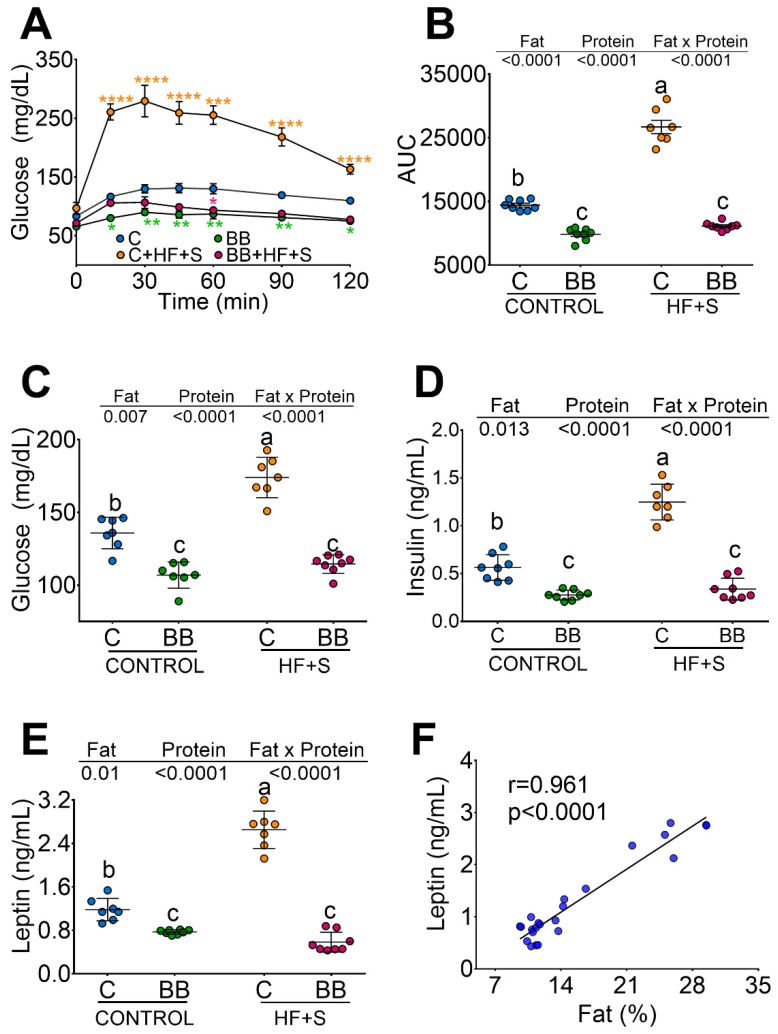
Black bean downregulates leptin and insulin concentration, decreasing glucose intolerance. (**A**) Glucose tolerance test after 2 months of dietary treatment with casein (**C**), black bean (BB), C + high-fat + 5%sucrose (C + HF + S), BB + HF + S. (**B**) Area under the curve (AUC), (**C**) fasting serum glucose, (**D**) fasting serum insulin, (**E**) fasting serum leptin, (**F**) correlation between leptin an fat mass, *n* = 8 per group.

**Figure 3 nutrients-12-01182-f003:**
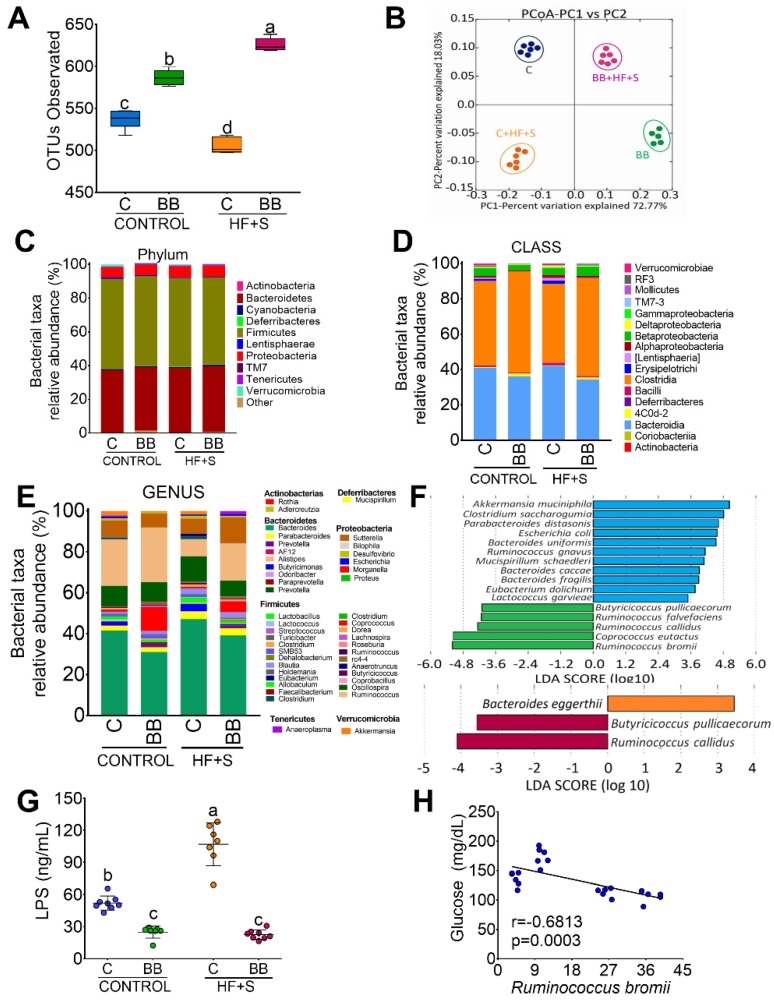
Black bean increases α-diversity and a cluster of bacterias of the Clostridia family and decreases metabolic endotoxemia and glucose concentration. (**A**) α-diversity after 2 months of dietary treatment with casein (**C**), black bean (BB), C + high-fat + 5% sucrose (C + HF + S), BB + HF + S. (**B**) Principal components analysis of the C, BB, C + HF + S, BB + HF + S groups, *n* = 6 per group. (**C**) relative abundance of the main phyla, (**D**) relative abundance of gut microbiota at the class level, (**E**) and at the genus level, (**F**) linear discriminat analysis (LDA), (**G**) serum LPS concentration, (**H**) correlation between *R. bromii* abundance and serum glucose concentration, *n* = 6 per group.

**Figure 4 nutrients-12-01182-f004:**
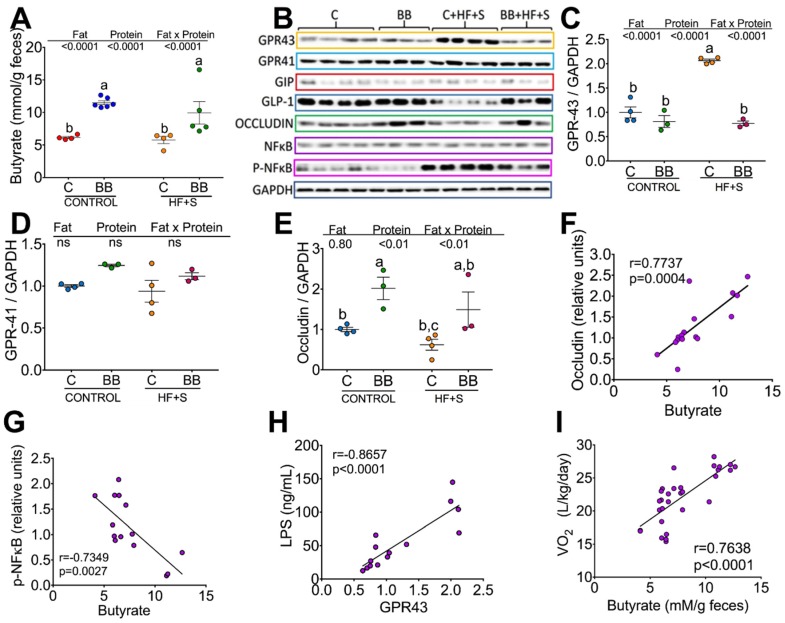
Consumption of black bean increases butyrate and decreases NF-κB protein abundance in the colon. (**A**) Fecal butyrate concentration, (**B**) Western blot analysis of G protein-coupled receptor (GPR) 41 and 43, occludin, NF-κB, P-NF-κB, (**C**–**E**) and densitometric analysis of GPR41, GPR43 and occludin after 2 months of dietary treatments with casein [C], black bean [BB], Casein + high-fat + S 5% [C + HF + S], BB + HF + S. (**F**) Correlation between occludin and butyrate, (**G**) correlation between NF-κB and butyrate, (**H**) correlation between LPS and GPR43. (**I**) correlation between VO2 and fecal butyrate concentration.

**Figure 5 nutrients-12-01182-f005:**
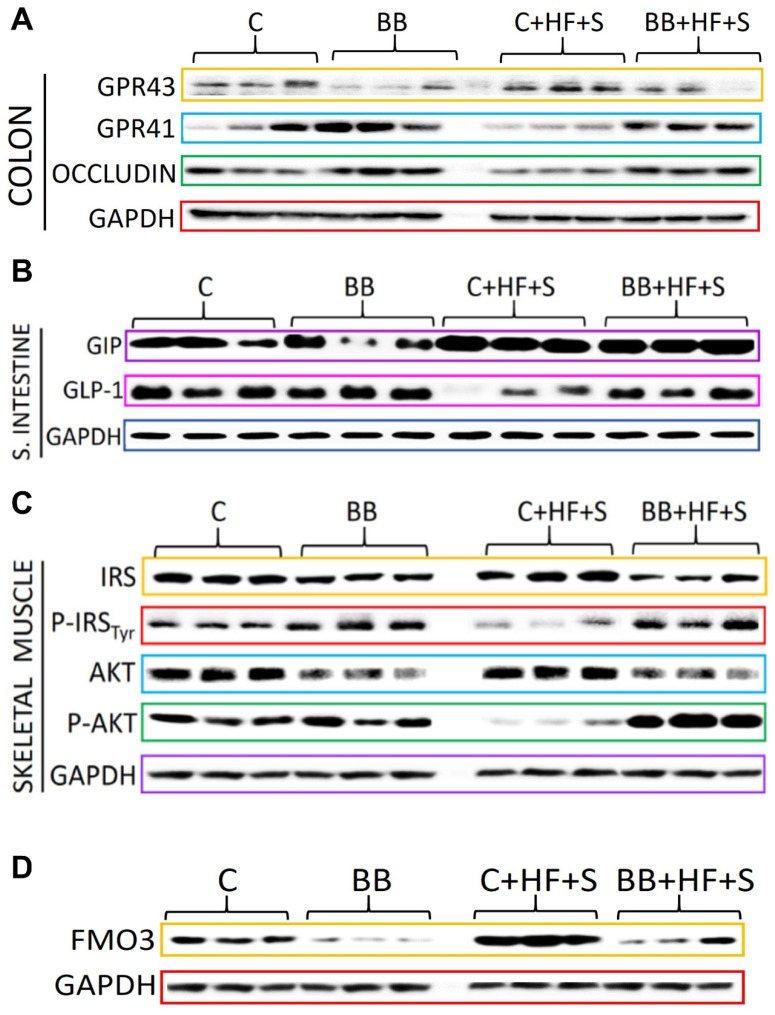
Black bean consumption maintains intestinal integrity and insulin signaling after an acute metabolic challenge with glucose. (**A**) Western blot analysis of small intestinal G-protein-coupled receptors (GPR43 and GPR41) and occludin. (**B**) Intestinal GIP, GLP-1, (**C**) skeletal muscle IRS, phosphorylation of IRS-1 at Tyr^896^, AKT, phosphorylation of AKT at Ser^473^ (**D**) Hepatic flavin monooxygenase 3 (FM03) after 2 months of dietary treatments with casein [C], black bean [BB], Casein + high-fat + S 5% [C + HF + S], BB + HF + S. Four rats of each group were injected with 2 g/kg body weight of glucose and killed 30 min later.

**Figure 6 nutrients-12-01182-f006:**
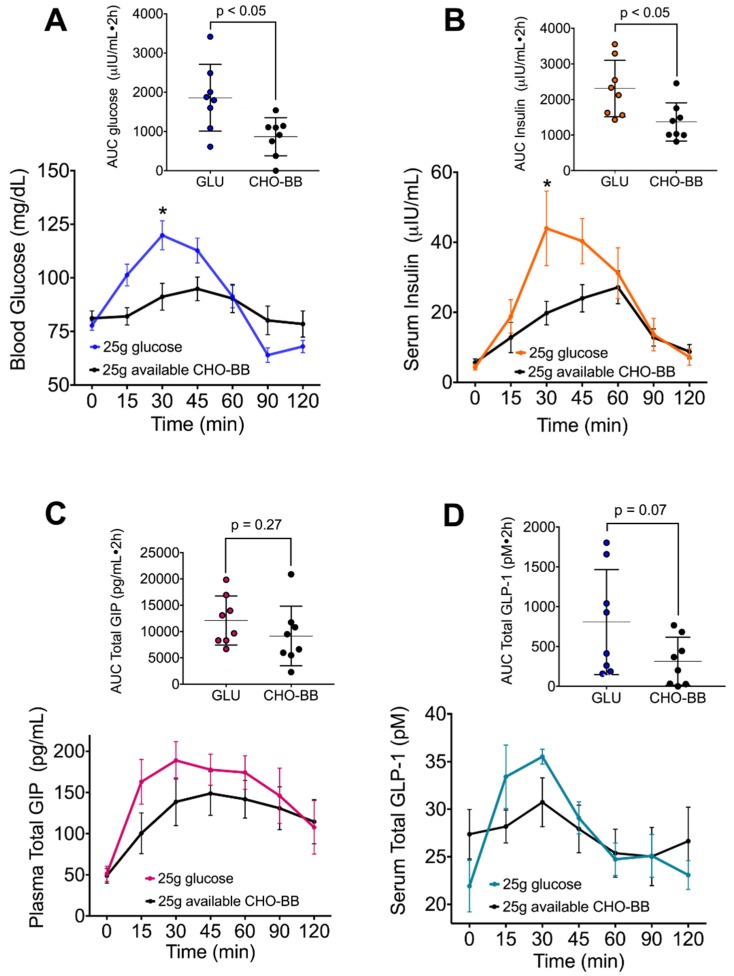
The consumption of 25 g of available carbohydrates from black bean attenuates serum glucose and postprandial insulin peaks. After consumption of 25 g of glucose or 25 g of available carbohydrates from black bean, blood samples were taken at different times over 2 h. (**A**) Serum glucose, (**B**) serum insulin, (**C**) total plasma GIP and (**D**) total serum GLP-1 in eight healthy subjects. For each graph, the area under the curve (AUC) was calculated and showed as an inset in each panel. Data are shown as mean ± SEM. Significant differences were considered with *p* < 0.005.

**Table 1 nutrients-12-01182-t001:** Composition of diets.

Ingredient (g/100 g)	C	BB	C + HF	BB + HF
Casein ^1^	17	-	24	-
Dry cooked black bean	-	79.8	-	78
Cornstarch	41	5.2	23.97	-
Dextrinized cornstarch	15	1.7	10.3	-
Sucrose	10	1.3	7.8	-
Soybean oil	7	7	7	-
Cellulose	5	-	5	-
Mineral mix ^2^	3.5	3.5	3.5	3.5
Vitamin mix ^3^	1	1	1	1
L-Cystine	0.3	0.3	0.3	0.3
Choline bitartrate	0.25	0.25	0.25	0.25
TBHQ	0.0014	0.0014	0.0014	0.0014
Lard	-	-	17	17
Sucrose in drinking water	-	-	5%	5%

^1^ Harland Teklad; ^2^ Mineral mix AIN-93-MX MP Bio; ^3^ Vitamin mix AIN-93-UBX MP Bio. TBHQ: tert-butylhydroquinone; C: casein; BB: black bean; C + HF: casein + high-fat diet; BB + HF: black bean + high-fat diet.

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
