# Peer review of "Consumption of Cooked Black Beans Stimulates a Cluster of Some Clostridia Class Bacteria Decreasing Inflammatory Response and Improving Insulin Sensitivity"

_nutrients, 2020, doi:10.3390/nu12041182_

Round 1

Reviewer 1 Report

The paper entitled “ consumption of resistant starch ……..” was reviewed, and this paper is interesting. However, there are some comments on it.

(1) In the method section, you should include the microbiota analysis. Did you use next generation analyzer of 16SDNA?
(2) The number of animals in each group is missing. In some experiments(Fig.1 and 2), N equals 8 per group, but in other experiments (Fig.3 4), N equals 6 and 4 per group. You should mention the reason why you used such different numbers of rats ? Did you select 4 rats in figure 4 from 6-8 rats in figures 1nad 3 ?
(3) You strengthen the importance of butylate production by BB. Are there significant differences of concentration of butylate among groups in figure 4A? It is well known that HFD decreased the SCFA in mice microbiota (Sasaki et al, Nutrients, 2019). In the current your paper, SCFAs in the HFD is higher than other groups. You need the comments in the discussion.
(4) In some beans, resistant protein is included and this protein plays important for microbiota (Tamura et al., Nutrients, 2019). Do black beans you used in this experiment contain such resistant protein?
(5) As you mentioned, the role of phenolic compounds on microbiota is well known. So please add more detail information in the discussion.

Author Response

We appreciated the time spent in the revision of the manuscript, your suggestions and comments.

Reviewer 1

  1. The microbiota analysis was included in the methods section
  2. In figure 1, the N=6 was due to the fact that the quality of the DNA in the faeces was not so good in some groups, so we decided to use an n=6

In the material and methods section, we modified the paragraph including the size of the experimental groups and the rationale of the study. We described that “each experimental group (n=8) was divided into two subgroups, the first subgroup was killed in fasting stage (n=4) and the other subgroup(n=4) had a metabolic challenge by intraperitoneal administration of a glucose

  1. Yes, the butyrate concentration was higher in the group fed black bean with respect to the control group. A panel with the fecal butyrate concentration was included instead of the total SCFA. A figure of total SCFA was in supplementary figure 1. However, the concentration of SCFA in the group fed HFD+BB was no significant different than the C group. I tried to upload the figure here, but it was no possible, but You can find it in Figure 4

Thank you very much for the reference. I read the article by Sasaki, et al, and the changes in SCFA were in cecal samples after a long fasting period. The different conditions and sampling site could modify the concentration of SCFA, However, I found a very interesting article indicating that after a high-fat diet, the stimulation of GPR43 receptor with SCFA leads to an increase in adipogenesis and impaired glucose control. The results are similar to those found in our study. We included a paragraph with new references in the discussion section as you suggested.

  1. Thank you for your comment, the resistant protein has been described in soybean, however, there is no information about the presence of resistant proteins in black beans. Further studies are needed on this topic.
  2. Ok, more detailed information was included on the role of phenolic compounds on gut microbiota in the discussion section.

Reviewer 2 Report

General comments

  1. This study provides some interesting information. However, the originality of the findings obtained in this study appears to be unclear or poor because several studies have been reported to show the similar effects of resistant starch on intestinal butyrate and microflora including butyrate-producing bacteria such as Clostridia in association with metabolic syndrome.
  2. The results indicating the lower hepatic FMO-3 abundance by BB are of interest. However, because of limited data, further development is necessary.

Specific comments

  1. Information of the composition of nutrients in dry cooked black bean used is not enough. Especially, the contents of dietary factors (dietary fibers, prebiotics, resistant starch, etc.) affecting microflora in the black bean should be indicated.
  2. There is no description for the statistical method of data analysis.
  3. There is no description for the methods of western blotting analysis and microbiota analysis.
  4. What is "heces" in Fig 4A?  What is "FAST" in Fig 4B?

Author Response

Reviewer 2

  1. Thank you for your comment. Several studies have been focused on resistant starch in bananas, potatoes or isolated resistant starch, however, billions of people around the world eat pulse or legumes as a source of protein and there is no information about resistant starch in black bean where the protein and the presence of resistant starch play an important role in modifying gut microbiota. We found an increase in some species belonging to class Clostridiales particularly Butyricicoccus pullicaecorum, Ruminicoccus flavefaciens, Ruminicoccus callidus and Ruminicoccus bromii after the consumption of black bean even in the presence of a high-fat diet. These species were significantly increased only in the BB group. These species were not associated with the consumption of a high-fat diet with casein.
  2. Yes, you are correct. The data about the decrease in the protein abundance of FMO3 after the consumption of BB is of interest because it has been demonstrated that the deletion of FMO3 conferred protection against obesity in mice (Cell Reports 19, 2451-2461, June 20, 2017). It has been demonstrated that the overexpression in the liver of mice of FMO3 leads to a significant increase in the TMAO levels (Cell Metabolism 17:49-60,2013). In fact, we observed that the group fed with BB showed less % of body fat. We included a supplementary figure in which we plotted % body fat vs protein abundance of FMO3 and we modified the paragraph with additional information.

Specific comments

  1. Information of the composition of nutrients in dry cooked black beans was added in a supplementary table.

2,3 Sorry, you are correct, I forgot to upload the methods. All methods are included

  1. The word “Heces” was changed to feces. We deleted the word FAST

Reviewer 3 Report

I have reviewed the paper by Sanchez-Tapia et al. Although the idea is interesting, the description of groups is not well-done, which affects the comparison and conclusions. There are several important findings that should be highlighted, making the paper more sound.

The Abstract does not indicate what AUC means.

The article need English editing, some sentences lack verbs.

The introduction, although well organized, it still lacks some details in the different RS groups (See https://digestivehealthinstitute.org/2013/05/10/resistant-starch-friend-or-foe/). This is important to provide a rationale for using dry cooked black beans.

Description of the groups is not clear. Authors state that “The first 2 groups were fed 17% casein protein (C), the second 69 group was fed 17% black bean protein from dry cooked blackkbean (BB). The other 2 groups were 70 fed a high fat diet (HF) diet and 5% sucrose (S) in drinking water with casein (C+HF+S) or black bean 71 protein source (BB+HF+S).”, but then in Table 1, we see that BB group is not receiving casein, and that BB+HF is not receiving sucrose (!).

Sucrose was not supposed to be in the diet of the C group, according to their description. Despite that they received the highest amount of it (Table 1) 10 vs 1.3. No surprise in seeing these rats gaining significantly more weight, and explain all the ‘unexpected” findings seen in Figure 1. This point is entirely obviated, and the authors without any basis move on to the issue of type of protein as the origin of the observed differences.

Results presented in 3.2 are due to administration of dry cooked black bean, not the resistant starch as they continuously assume. Agaiin here (Figure 2) higher glucose can be explain by the increased amount of sucrose in the C and C+HF+S groups.

Title of the paper talks about Clostridia class, however not Class data is presented in Figure 3. At the genus level, what is most striking from Figure 3D is the increased abundance of Rothia (in groups with BB), and increased in Prevotella in C+HF+S. Clostridium saccharogumia is absent in BB treated groups Figure 3F, but exhibit a great amount of Coprococcus eutactus and Ruminococcus bromii, which are Class Gammaproteobacteria, and Clostridia, respectively.. Authors point out that these are butyrate producers, but it is unclear how the authors then make the statement “As a consequence, we showed that the BB 244 groups had the lowest circulating LPS concentration”.How an increase in representation of certain Clostridiales, decreases serum LPS???

Results from 3.4 are unexpected, as the Introduction explains that upon entry into the colon RS will yield SCFA (the acronym should have been placed there, by the way). Despite this, BB groups did not show the highest SCFA.

After presenting Figure 4, authors go back to LPS in Figure 3G. Suggest moving 3G into Figure 4. Most importantly, why are we looking for a correlation between LPS and GPR43, what is the rationale? The phrase “High fat diet increased LPS concentration associated with a high intestinal permeability.” Needs to the rewritten.

Honestly, the finding in Figure 4G might be the most important of all the study, and authors may want to consider changing the title and the flow of the paper.

Author Response

The description of the groups was modified for a better understanding of the study in the materials and methods section.

In the abstract section,  the meaning of the AUC was described

The manuscript was sent to editing services

The introduction was modified as suggested by the reviewer and we included details of the different resistant starch and we included 2 references

A more detailed description of the groups and the rationale of the study were included in the material and methods section as suggested by the reviewer

Thank you for your observations, we considered that the confusion in the interpretation of the results were due to a incomplete description of the dietary treatments and diets. We modified this section, and table 1 was modified to include the sugar in the drinking water to be more clear.

A figure of the class level of the gut microbiota was included in Figure 3 (3D) and figure legend was modified.

Gram (-) bacteria contains in its membrane cell a lipopolysaccharide associated with an increase in circulating levels of LPS. Previous studies in humans and animals have demonstrated that an increase in Gram (-) bacteria increase circulating levels of LPS. We observed that the groups fed C+HFD+S had a significant increase in Bacteroides eggerthii that is a Gram (-) bacteria. This group showed the highest level of serum LPS. Whereas the group fed BB+HF+S showed a significant increase in 3 Gram (+) bacteria and 1 Gram (-) bacteria 

We use the LDA analysis to identify the bacteria species that were significantly modified in the gut microbiota considering the relative abundance of each species and the sample size, for these reason we included this plot indicating the significant differences in the type of bacteria in each group.

You are correct the BB did not show the highest total SCFA, but the BB groups showed the highest fecal butyrate concentration, for this reason, we included a panel in figure 4 with fecal butyrate concentration (4A) because we considered this result more relevant than the Total SCFA  that was included in supplementary figure 1.

Because the LPS is more associated with gut microbiota, we considered that serum LPS should remain in Figure 3.

We agree with the reviewer that the negative correlation between NF-kB-P and fecal butyrate concentration is an interesting finding, however, we would like to emphasize to that consumption of black bean, a food that plays an important role as a source of protein, and resistant starch in the diet of many countries. For us, it was a very important finding that the group of rats that fed BB gained less body fat and had the same lean mass as the groups fed casein. Our study shows, that the consumption of BB has a beneficial effect on gut microbiota, and some of the health benefits can be now attributed to this modification of gut microbiota even after consuming a high-fat diet. In addition, preliminary results in our lab demonstrated that consumption of a BB protein concentrate which does not contains resistant starch showed that gut microbiota was different than the dry cooked bean, suggesting that the presence of resistant starch is important in the modification of gut microbiota mainly in the Clostridia class. We modified the title of the manuscript.

Round 2

Reviewer 2 Report

The revised manuscript appears to appropriate for the publication. 

Author Response

Thank you.

Reviewer 3 Report

Thank you to the authors for providing their revised version so promptly.

It is now clear that from the new improved Table 1, that the groups are indeed Not Comparable, judging from the amount of sucrose provided to

C BB C+HF BB+HF

Sucrose 10 1.3 7.8 -

Just this factor by itself could explain the differences in Figure 2.

I am uncertain what authors mean by “Because the LPS is more associated with gut microbiota” in their rebuttal. Even their graph in Figure 3C show that there are more Firmicutes (Gram positive) than Bacteroidetes (Gram negative)

Authors state in their rebuttal that:

You are correct the BB did not show the highest total SCFA, but the BB groups showed the highest fecal butyrate concentration, for this reason, we included a panel in figure 4 with fecal butyrate concentration (4A) because we considered this result more relevant than the Total SCFA that was included in supplementary figure 1.

I just mentioned the result was unexpected. The value of the findings rely on the correlation with butyrate, and that is why I suggested it to be the main result to be highlighted.

Despite this the discussion keeps bringin SCFA rather than butyrate. Statements such as “Fermentation of these substrates provides SCFA, mainly butyrate, that is beneficial to the health of the colon …” although true, need a valid reference (no the one provided [17].

Author Response

Reviewer 3.

Thank you for your comments. In order to clarify how the diets were designed, I am sending you an explanation and a table to see the macronutrient and energy content of each diet. Diets were designed according to the American Institute of Nutrition. The percent of protein in this study was 17%, and we used casein as the reference protein. Then, we adjusted the same amount of protein using the dry cooked black bean that contained 17.82 g of protein/100g of dry cooked beans. We then adjusted the control and the dry cooked bean diets to the same percentage of lipids, carbohydrates and dietary fiber, as well as minerals and vitamins according to the AIN93.

Energy and macronutrient content in the diets

Nutrient

C

BB

HF

BB+HF

Carbohydrates

66.0

66.3

42.1

42.5

Lipids

7.0

7.0

24.0

23.9

Protein

17.0

16.9

24.0

23.9

kcal/g diet

3.95

3.96

4.80

4.81

In order to make a high-fat diet, we increased the percentage of lipids by 2.4-fold compared to the control AIN93 diet, resulting in approximately 24% of lipids compared to 7% of lipids in the control diet.  As a result, control or BB diets contained approximately 3.95 Kcal/g, whereas the high-fat diets (C+HF+S or BB+HF+S) contained approximately 4.8 Kcal/g.

There was no significant difference at the phylum level between Firmicutes and Bacteroidetes in any group

The percentages were:

Control group= Bacteroidetes 37.1%, Firmicutes 53.4%

BB group= Bacteroidetes 37.6%, Firmicutes 53.1%

C+HFD+S = Bacteroidetes 38.3%, Firmicutes 52.5%

BB+HFD+S= Bacteroidetes 39%, Firmicutes 51.7%

It has been reported that although most Firmicutes are Gram(+) they also contain Gram(-) bacteria. (Lock, B.A. et al Adv in Applied Microbiology, 2019).

Thank you for the suggestion, we included 3 references

Baxter, N.T.; Schmidt, A.W.; Venkataraman, A.; Kim, K.S.; Waldron, C.; Schmidt, T.M. Dynamics of Human Gut Microbiota and Short-Chain Fatty Acids in Response to Dietary Interventions with Three Fermentable Fibers. mBio 2019, 10, doi:10.1128/mBio.02566-18.

Donohoe, D.R.; Garge, N.; Zhang, X.; Sun, W.; O'Connell, T.M.; Bunger, M.K.; Bultman, S.J. The microbiome and butyrate regulate energy metabolism and autophagy in the mammalian colon. Cell Metab 2011, 13, 517-526, doi:10.1016/j.cmet.2011.02.018.

Bach Knudsen, K.E.; Laerke, H.N.; Hedemann, M.S.; Nielsen, T.S.; Ingerslev, A.K.; Gundelund Nielsen, D.S.; Theil, P.K.; Purup, S.; Hald, S.; Schioldan, A.G., et al. Impact of Diet-Modulated Butyrate Production on Intestinal Barrier Function and Inflammation. Nutrients 2018, 10, doi:10.3390/nu10101499